# Epipharyngeal Abrasive Therapy (EAT) Has Potential as a Novel Method for Long COVID Treatment

**DOI:** 10.3390/v14050907

**Published:** 2022-04-27

**Authors:** Kazuaki Imai, Takafumi Yamano, Soichiro Nishi, Ryushiro Nishi, Tatsuro Nishi, Hiroaki Tanaka, Toshiyuki Tsunoda, Shohei Yoshimoto, Ayaki Tanaka, Kenji Hiromatsu, Senji Shirasawa, Takashi Nakagawa, Kensuke Nishi

**Affiliations:** 1Mirai Clinic, Fukuoka 812-0013, Japan; imakazu@mirai-iryou.com; 2Section of Otolaryngology, Department of Medicine, Fukuoka Dental College, Fukuoka 814-0193, Japan; yamano@college.fdcnet.ac.jp; 3Nishi Otolaryngology Clinic, Fukuoka 814-0031, Japan; nishi2416@outlook.jp (S.N.); ryushiro0324@icloud.com (R.N.); t.nishi.ir@adm.fukuoka-u.ac.jp (T.N.); 4Department of Otolaryngology, Faculty of Medicine, Fukuoka University, Fukuoka 814-0180, Japan; 5Tanaka Hiroaki Clinic, Fukuoka 814-0142, Japan; amdb9@goo.jp; 6Department of Cell Biology, Faculty of Medicine, Fukuoka University, Fukuoka 814-0180, Japan; tsunoda@fukuoka-u.ac.jp (T.T.); sshirasa@fukuoka-u.ac.jp (S.S.); 7Section of Pathology, Department of Morphological Biology, Division of Biomedical Sciences, Fukuoka Dental College, Fukuoka 814-0193, Japan; yoshimoto@college.fdcnet.ac.jp; 8Tanaka ENT Clinic, Osaka 553-0006, Japan; tuvajp@gmail.com; 9Department of Microbiology and Immunology, Faculty of Medicine, Fukuoka University, Fukuoka 814-0180, Japan; khiromatsu@fukuoka-u.ac.jp; 10Department of Otorhinolaryngology, Graduate School of Medical Sciences, Kyushu University, Fukuoka 812-8582, Japan; nakagawa.takashi.284@m.kyushu-u.ac.jp

**Keywords:** long COVID, epipharyngeal abrasive therapy (EAT), myalgic encephalomyelitis/chronic fatigue syndrome (ME/CFS), chronic epipharyngitis

## Abstract

COVID-19 often causes sequelae after initial recovery, referred to collectively as long COVID. Long COVID is considered to be caused by the persistence of chronic inflammation after acute COVID-19 infection. We found that all long COVID patients had residual inflammation in the epipharynx, an important site of coronavirus replication, and some long COVID symptoms are similar to those associated with chronic epipharyngitis. Epipharyngeal abrasive therapy (EAT) is a treatment for chronic epipharyngitis in Japan that involves applying zinc chloride as an anti-inflammatory agent to the epipharyngeal mucosa. In this study, we evaluated the efficacy of EAT for the treatment of long COVID. The subjects in this study were 58 patients with long COVID who were treated with EAT in the outpatient department once a week for one month (mean age = 38.4 ± 12.9 years). The intensities of fatigue, headache, and attention disorder, which are reported as frequent symptoms of long COVID, were assessed before and after EAT using the visual analog scale (VAS). EAT reduced inflammation in the epipharynx and significantly improved the intensity of fatigue, headache, and attention disorder, which may be related to myalgic encephalomyelitis/chronic fatigue syndrome (ME/CFS). These results suggest that EAT has potential as a novel method for long COVID treatment.

## 1. Introduction

Long COVID refers to a series of health consequences that are present four or more weeks after infection with SARS-CoV-2 [1,2]. A systematic review and meta-analysis revealed that 80% of patients developed one or more long-term symptoms, including fatigue, headache, attention disorder, hair loss, sore throat, and dyspnea [3]. Myalgic encephalomyelitis/chronic fatigue syndrome (ME/CFS) is a frequently mentioned symptom of long COVID [4]. Hyperinflammation due to COVID-19 can cause ME/CFS, but the causal organs associated with ME/CFS have not been identified and no standardized treatment has been developed [4,5,6,7,8]. The epipharynx is a primary target for SARS-CoV-2 replication in the early stages of COVID-19 [9]. The activated lymphoid tissue of the epipharynx produces various inflammatory cytokines for the clearance of pathogens and this immune activation is considered to be one of the pathologies of ME/CFS [10,11,12]. A relationship has already been established between chronic epipharyngitis and ME/CFS, and epipharyngeal abrasive therapy (EAT), a treatment for chronic epipharyngitis in Japan, is reported to be effective against ME/CFS [9,13,14,15]. In this study, we endoscopically confirmed that all patients with long COVID had residual inflammation of the epipharynx. To examine whether EAT is an effective treatment for long COVID, which has symptoms similar to those of ME/CFS, we evaluated the intensity of symptoms, including fatigue, headache, and attention disorder, before and after EAT using the visual analog scale. We found that continuous EAT significantly improved the intensity of the symptoms of long COVID via the reduction of chronic epipharyngitis and neuromodulation by stimulation of the vagus nerve ending in the epipharynx. It was reported that about 70% of patients had sequelae even after half a year of acute infection but EAT relieved the symptoms of long COVID in one month [16]. Our results suggest that EAT may be an effective treatment for long COVID.

## 2. Materials and Methods

### 2.1. Patients

The study period was from August 2021 to November 2021. The population for this study consisted of 58 consecutive COVID-19 patients who visited the specialty outpatient clinic for long COVID, aged 13 to 68 years (mean age: 38.4 ± 12.9 years), with a confirmed diagnosis of SARS-CoV-2 by real-time reverse transcriptase polymerase chain reaction (RT-PCR) and long COVID symptoms, including fatigue, headache, and attention disorder, for more than 2 months (mean: 113.6 ± 76.4 days, median: 81 days). Among the patients, 25 and 14 were aware of throat pain and post-nasal drip, respectively, that is, more than half of the patients had no throat symptoms. All the participants provided informed written consent. This research was conducted in accordance with the Declaration of Helsinki and Title 45, US Code of Federal Regulations, Part 46, Protection of Human Subjects, effective as of 13 December 2001.

### 2.2. Epipharyngeal Abrasive Therapy (EAT)

A 0.5% ZnCl_2_ solution was applied to the epipharyngeal mucosa of patients in the outpatient department once a week using a sterile straight nasal cotton swab and a pharyngeal swab (Figure 1). To increase the effectiveness of EAT, it is important to thoroughly scrub the epipharyngeal wall with cotton swabs. ZnCl_2_ solution is commonly used as a chemical solution for EAT [9,13,15,17].

### 2.3. Diagnosis and Grading of Chronic Epipharyngitis

All the patients underwent an endoscopy at the first visit and following one month of treatment with EAT. We diagnosed and graded chronic epipharyngitis based on these endoscopic findings. The inflammation grade of chronic epipharyngitis was scored on a scale of 0–6 (0: absent, 1–2: mild inflammation, 3–4: moderate inflammation, 5–6: severe inflammation) (Figure 2).

### 2.4. Efficacy Analysis of EAT for Chronic Epipharyngitis and Long COVID

Efficacy analyses were performed for patients using the inflammatory grade of chronic epipharyngitis and a visual analog scale (VAS) for the severity of fatigue, headache, and attention disorders (0: absence of symptoms, 10: highest severity of symptoms) before and after EAT for 1 month. Statistical analyses were performed using paired or unpaired two-tailed Student’s *t*-tests. All *p*-values less than 0.05 were considered statistically significant. Wilcoxon signed-rank testing was conducted and reported alongside corresponding effect sizes for EAT (Cohen’s d; classified as small (0.2), medium (0.5), large (0.8), or very large (1.2)).

## 3. Results

### 3.1. EAT Reduced Inflammation in the Epipharynx

All patients with long COVID had chronic epipharyngitis and were treated with EAT once a week for one month. EAT resulted in reduced redness, cobblestone-like granular changes in the epipharyngeal mucosa, and a decrease in the degree of impure bleeding during EAT, which correlates with the intensity of inflammation (Figure 3a). EAT yielded a statistically significant improvement in the inflammatory grade (*p* = 5.4 × 10^−7^, d = −0.92 (large); Table 1). These results suggest that EAT reduces the residual inflammation of the epipharynx in patients with long COVID.

### 3.2. The Severity of Long COVID Symptoms Decreased Due to EAT

The therapeutic efficacy of EAT was assessed by comparing the visual analog scale (VAS) scores for fatigue, headache, and attention disorder pre-EAT treatment (before) and following one month of treatment with EAT (after) (Figure 4). Before treatment, the VAS score for fatigue was the highest among the three symptoms (Table 1). The mean VAS scores for fatigue, headache, and attention disorder following one month of treatment with EAT (after) were 1.61-fold, 1.93-fold, and 1.48-fold lower than those observed pre-EAT treatment (before), respectively (fatigue: *p* = 1.6 × 10^−7^, d = −0.66 (medium), headache: *p* = 5.7 × 10^−7^, d = −0.76 (medium), attention disorder: *p* = 1.6 × 10^−7^, d = −0.58 (medium); Table 1). These results suggest that fatigue is the most severe symptom of long COVID and that EAT improves the severity of long COVID symptoms. Furthermore, to clarify whether the improvements in long COVID symptoms by EAT were due to the reductions of epipharyngitis, improvements in VAS scores were analyzed between 35 patients with an inflammatory grade improvement of 1 or more and 23 patients without improvement. The mean improvements in VAS scores for fatigue, headache, and attention disorder in patients with reductions in epipharyngitis were 2.24-fold, 1.10-fold, and 1.20-fold higher than those in patients without reductions in epipharyngitis, respectively (fatigue: *p* = 0.04, headache: *p* = 0.79, attention disorder: *p* = 0.72; Figure 4 and Figure 5). These results suggest that reductions in epipharyngitis by EAT are most strongly involved in fatigue improvement.

## 4. Discussion

Fatigue is the most prevalent symptom of the various sequelae of long COVID, similar to myalgic encephalomyelitis/chronic fatigue syndrome (ME/CFS) [18,19,20]. The Centers for Disease Control and Prevention (CDC) defines ME/CFS as a condition with severe fatigue that lasts for at least 6 months with at least four of the following symptoms: cognitive impairment, tender cervical or axillary lymph nodes, headache, muscle and multi-joint pain, sore throat, post-exertional malaise (PEM), and unrefreshing sleep [21]. Though the pathogenesis of ME/CFS is not well understood, many cases of ME/CFS are considered to be triggered by acute infection, including upper respiratory tract infection, and inflammation states after infection may be implicated [18,22,23,24]. Similarly, long COVID is caused by SARS-CoV-2, which is transmitted through the upper respiratory tract, meaning that ME/CFS and long COVID may share certain underlying pathological processes [25,26]. In fact, it has been indicated that 14.3% of long COVID patients who experienced fatigue for more than 6 months met the ME/CFS criteria [27]. However, although hyperbaric oxygen therapy, vitamin C injection, and the oral administration of adaptogens are reported to be effective treatments for fatigue caused by long COVID, no standardized treatments have been reported at this time [28,29,30,31].

In this study, we performed endoscopy on all 58 patients who visited the specialty outpatient clinic for long COVID and found that all had chronic epipharyngitis. This study is the first report to show that chronic epipharyngitis is present in patients with long COVID regardless of the presence or absence of throat symptoms and that it may be involved in the pathophysiology of long COVID. Located at the back of the nasal cavities, the epipharynx, with its high expression of SARS-CoV-2 entry factors, is a primary target for SARS-CoV-2 replication in the early stages of COVID-19 [9]. Similar to COVID 19, upper respiratory tract infections (URTIs) of viral etiology begin in the epipharynx [32,33]. Chronic epipharyngitis, which may be secondary to URTI, is considered a residual immune response and causes not only various upper respiratory tract symptoms, such as postnasal drip, throat pain, and cough, but also systemic symptoms, such as chronic fatigue, headache, systemic pain, and dizziness [12,13,31,32]. Some of these systemic symptoms resemble those of MS/CFS, and chronic epipharyngitis is reported to activate epipharyngeal inner immunity and produce neuroexcitatory molecules, such as TNF-α, IFN-γ, and IL-1, which activate the immune system of the brain, resulting in low-grade inflammation in the brain associated with MF/CFS [12,13,34,35,36,37]. In this study, we found that 90% of long COVID patients had moderate or severe chronic epipharyngitis and that all patients complained of either fatigue, headache, attention disorder, or a combination of the three, confirming the findings of other reports on long COVID (Figure 3 and Figure 4). These results suggest that ME/CFS-like symptoms seen in patients with long COVID may be caused by residual inflammation of the epipharynx after SARS-CoV-2 infection. The diagnostic criteria for ME/CFS include a sore throat that can occur with chronic epipharyngitis, which may support our results [21].

Epipharyngeal abrasive therapy (EAT) was originally developed in Japan and is the only treatment known to be effective for chronic epipharyngitis [9,13]. The mechanisms of EAT are reported to be the anti-inflammatory effect of ZnCl_2_, the blood-letting effect, and vagus nerve stimulation [38]. Confirmation of the epipharyngeal mucosa with an endoscope is effective for diagnosing chronic epipharyngitis [15]. In non-inflamed epipharyngeal mucosa, the vasculature can be clearly seen in endoscopic photographs, while the vasculature is not apparent owing to submucosal edema and congestion in the inflamed epipharyngeal mucosa [39]. We assessed the inflammation scores of patients using their endoscopic characteristics and impure bleeding during EAT, which correlates with the intensity of inflammation (Figure 2). In this study, all 58 patients were treated with EAT after consenting to treatment. We reported that EAT significantly improved chronic epipharyngitis, as has previously been reported [9,17], and improved the major symptoms of long COVID. The reduction in chronic epipharyngitis due to EAT contributed most strongly to the improvement of fatigue. Interestingly, even in patients whose inflammation did not improve, EAT alleviated the three symptoms of fatigue, headache, and attention disorder. These results indicate that EAT is effective at improving long COVID symptoms through a complex mechanism. Regarding the reduction of mucositis, we previously showed that continuous EAT induces squamous metaplasia and submucosal fibrosis and suppresses the focal aggregation of inflammatory cells in the epipharyngeal mucosa [9,40]. These histological changes are also observed with laser treatment aimed at suppressing inflammation in allergic rhinitis [41,42]. We suggest that decreased inflammatory cytokine secretion, associated with the suppression of inflammatory cell aggregation in the epipharynx caused by EAT, improves the underlying pathology of ME/CFS. Another speculated mechanism of EAT with respect to long COVID is neuromodulation by stimulation of the vagus nerve ending in the epipharynx [38]. Vagus nerve stimulation (VNS) has been shown to suppress inflammatory cytokine levels that cause various systemic symptoms via activation of the cholinergic anti-inflammatory pathway [43,44,45]. Actually, non-invasive VNS is considered to be useful to modulate peripheral and central inflammatory response, headache, and mental distress associated with COVID-19 [46,47]. In other words, EAT is a treatment that can directly stimulate the vagus nerve without electrical devices [38], and this effect may have improved symptoms even in patients without reduced inflammation in the epipharynx.

Even though it has been reported that 87% of patients have sequalae at 2 months after acute infection and 76% at 6 months [16,48], we revealed the effectiveness of EAT for long COVID in a one-month study period. EAT is generally performed against chronic epipharyngitis for several months [9], thus further continuous EAT may give better results.

It should be noted that no control group was defined in this study and that the patients’ complaints of fatigue were of shorter duration than the ME/CSF definition of 6 months. Despite this, we have shown that EAT has the potential to improve the major symptoms associated with ME/CFS in long COVID patients.

## 5. Conclusions

Although pharyngeal symptoms may be clinically asymptomatic or mild, chronic epipharyngitis is frequently present in patients with long COVID. EAT may be a simple method of treatment for long COVID.

## Figures and Tables

**Figure 1 viruses-14-00907-f001:**
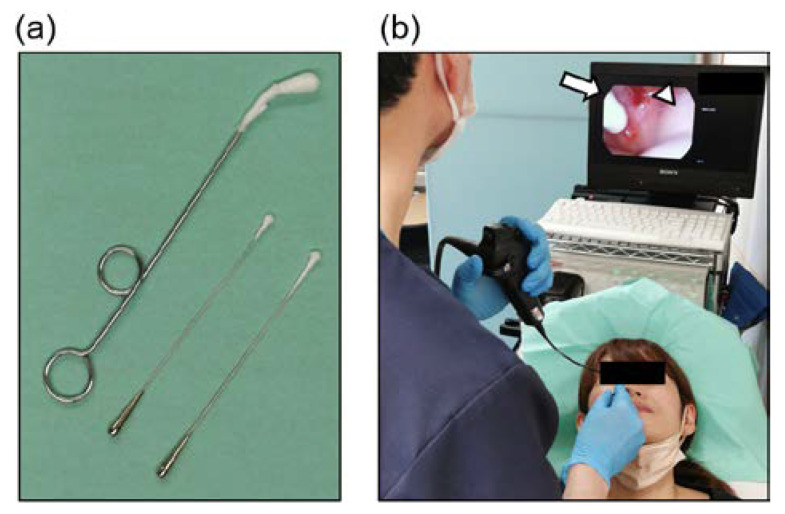
Epipharyngeal abrasive therapy (EAT). (**a**) Cotton swabs for EAT. (**b**) The method of endoscopic EAT. The entire epipharyngeal wall is scrubbed using a sterile straight nasal cotton swab soaked in 0.5% ZnCl_2_ solution. The white arrow indicates a sterile straight nasal cotton swab. The white triangle indicates impure bleeding.

**Figure 2 viruses-14-00907-f002:**
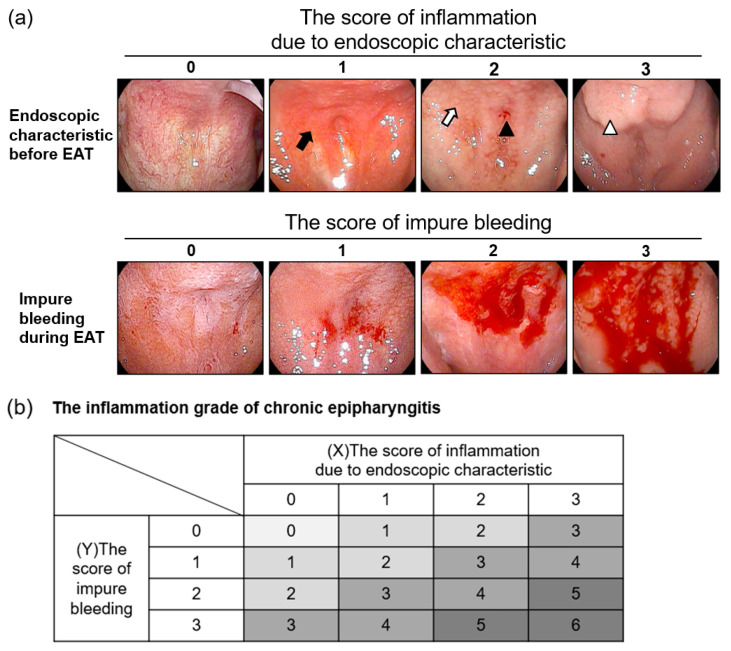
The inflammation grade of chronic epipharyngitis. (**a**) The endoscopic findings of chronic epipharyngitis. The upper panels show the endoscopic characteristics before epipharyngeal abrasive therapy (EAT). The score of inflammation was assessed using endoscopic characteristics. The black arrow indicates redness of the mucosa. The white arrow indicates cobblestone-like granular changes. The black triangle indicates submucosal bleeding. The white triangle indicates severe mucosal swelling. The lower panels show impure bleeding during EAT, which correlates with the intensity of inflammation. The score of impure bleeding was evaluated by the degree of bleeding. (**b**) The inflammation grade of chronic epipharyngitis was calculated using (X) the score of inflammation due to endoscopic characteristics and (Y) the score of impure bleeding. The inflammation grade of chronic epipharyngitis was scored on a scale of 0–6 (0: absent, 1–2: mild inflammation, 3–4: moderate inflammation, 5–6: severe inflammation).

**Figure 3 viruses-14-00907-f003:**
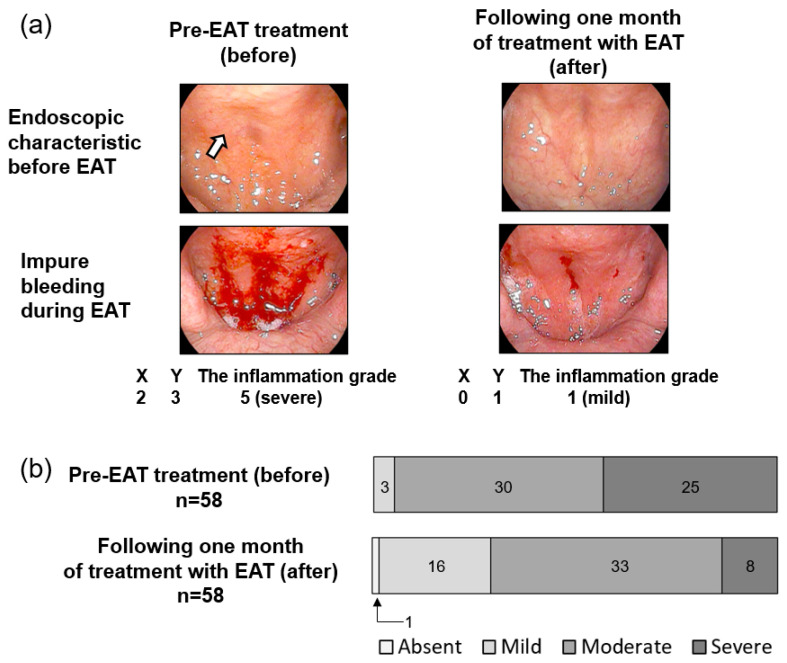
Effect of epipharyngeal abrasive therapy (EAT) for the treatment of chronic epipharyngitis. (**a**) Trans-nasal endoscopic photographs of the epipharynx in a patient. Left panel shows the epipharynx pre-EAT treatment (before). Right panel shows the epipharynx following one month of treatment with EAT (after). The white arrow indicates cobblestone-like granular changes. The inflammation grade = (X) the score of inflammation due to endoscopic characteristic + (Y) the score of impure bleeding. (**b**) Changes in inflammation grade due to EAT. All patients were treated with EAT once a week. The upper panel shows the inflammation grade at the first visit. The lower panel shows the inflammation grade following one month of treatment with EAT. The inflammation grade of chronic epipharyngitis was scored on a scale of 0–6 (0: absent, 1–2: mild inflammation, 3–4: moderate inflammation, 5–6: severe inflammation).

**Figure 4 viruses-14-00907-f004:**
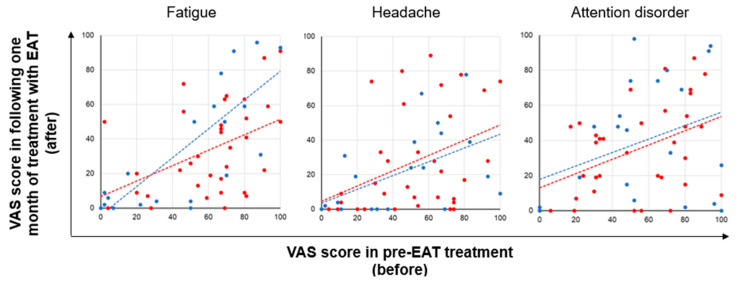
Visual analog scale (VAS) scores pre-epipharyngeal abrasive therapy (EAT) treatment (before) and following one month of treatment with EAT (after). Red circles indicate 35 patients with an inflammatory grade improvement of 1 or more. Blue circles indicate 23 patients without an inflammatory grade improvement. Dashed lines are added to guide the eye.

**Figure 5 viruses-14-00907-f005:**
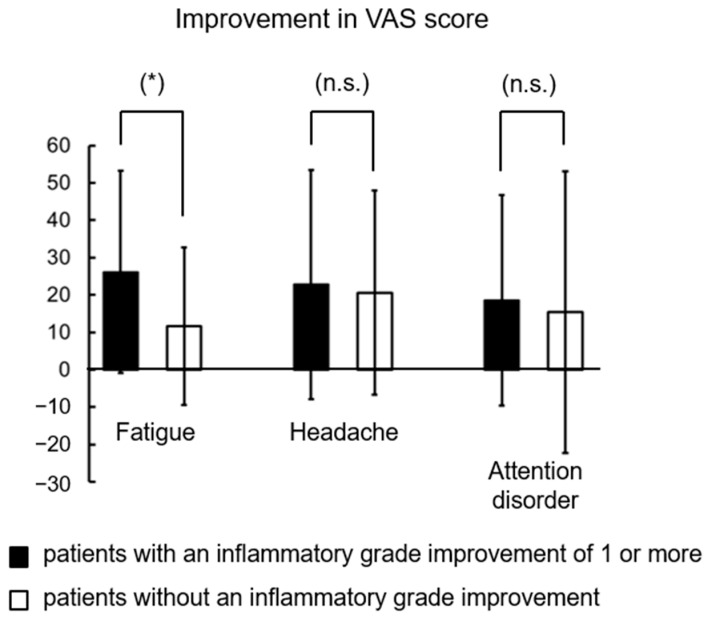
Improvements in VAS scores in 35 patients with an inflammatory grade improvement of 1 or more and 23 patients without an inflammatory grade improvement. Parametric unpaired *t*-test. * Significantly different at *p* < 0.05. n.s.: not significantly different.

**Table 1 viruses-14-00907-t001:** The significance of the difference and the effect sizes between pre-epipharyngeal abrasive therapy (EAT) treatment (before) and following one month of treatment with EAT (after).

	Pre-EAT Treatment (before)			Following One Month of Treatment with EAT (after)			Significance of Difference	Effect Sizes
Parameters	Mean	Median	SD	Mean	Median	SD	*p*-Value ^a^	Cohen’s d ^b^
Inflammation grade	4.69		1.13	3.57		1.31	5.4 × 10^−7^ *	−0.92
VAS score for fatigue	53.64	65	31.54	33.24	23	30.24	1.6 × 10^−7^ *	−0.66
VAS score for headache	45.48	49	30.05	23.57	11	27.38	5.7 × 10^−7^ *	−0.76
VAS score for attention disorder	53.43	52	29.79	36.12	36	29.66	1.6 × 10^−4^ *	−0.58

^a^ Parametric paired *t*-test. * Significantly different at *p* < 0.05. ^b^ Classified as small (0.2), medium (0.5), large (0.8), or very large (1.2).

## Data Availability

Not applicable.

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
