# Peer review of "Epipharyngeal Abrasive Therapy (EAT) Has Potential as a Novel Method for Long COVID Treatment"

_viruses, 2022, doi:10.3390/v14050907_

Round 1
Reviewer 1 Report
In this study, Imai and coworkers reported the universal existence of epipharyngitis in patients suffering from long COVID symptoms. They further found that epipharyngeal abrasive therapy (EAT), the therapy for epipharyngitis, significantly improved the essential symptoms of long COVID in association with resolution of epipharyngitis. Considering the facts that no established treatment for long COVID symptoms exist for now, the prevalence of COVID-19 is still high in the world and therefore there are many patients suffering from long COVID symptoms, the findings of this study is very important in the field of medical science. The concept of the study, its design, the statistical methods, results and the discussions are generally well written. However, as described below, there are several points that should be clarified or modified.
Major comment
- Essentially, the effect of EAT should be evaluated by comparing patients of long COVID with EAT-treatment and those without EAT-treatment. However, as the authors indicated, there are no specific control groups in this study, which is unavoidable matter. Therefore, the authors should precisely describe the reported natural course of long COVID symptoms in general, so that the readers could image the relative condition for those with treatment and without treatment. The impact of the present finding would change considerably whether the symptoms continue unchanged over years or are relieved and diminished within several months as natural course.
- Somewhat related to the above point, the timing of EAT induction after the onset or the diagnosis of COVID-19 infection should be clearly presented. Because it would offer different implications whether the treatment is introduced just one month after the onset of COVID-19 infection or more than 1 year after the onset of infection. There should be a certain level of naturally improvement tendency at the earlier disease phase of long COVID, so the timing of treatment induction in the disease course would be very important.
Minor comment
In the Methods section, the authors described that the diagnosis method for COVID-19 just as “positive SARS-CoV-2 test”. To provide the information on the accuracy of the diagnosis for COVID-19, the authors should present whether the test is for the existence of SARS-CoV-2 antigen or SARS-CoV-2 mRNA by real time RT-PCR.
Author Response
Dear Reviewer 1
Thank you very much for reviewing our manuscript and offering valuable advice.
We have addressed your comments with point-by-point responses, and revised the manuscript accordingly.
Response to Reviewer 1 Comments
Major Point 1: Essentially, the effect of EAT should be evaluated by comparing patients of long COVID with EAT-treatment and those without EAT-treatment. However, as the authors indicated, there are no specific control groups in this study, which is unavoidable matter. Therefore, the authors should precisely describe the reported natural course of long COVID symptoms in general, so that the readers could image the relative condition for those with treatment and without treatment. The impact of the present finding would change considerably whether the symptoms continue unchanged over years or are relieved and diminished within several months as natural course.
Major Response 1: Thank you very much for your excellent suggestion. Studies around the world have reported various incidence rates for long covid with different follow-up examination times after the acute infection, including 76% of people at 6 months, 87% at 60 days, and 96% at 90 days. These finding show that a substantial proportion of people who have had COVID-19 may develop Long COVID. Although there are no reports of patient follow-up for more than a year, Patients can be affected by long-term Long COVID for more than half a year. We have added text to the Introduction and Discussion sections (Page 2, Lines 16-18; Page 7, Lines 50- Page 8, Lines 2).
Major Point 2: Somewhat related to the above point, the timing of EAT induction after the onset or the diagnosis of COVID-19 infection should be clearly presented. Because it would offer different implications whether the treatment is introduced just one month after the onset of COVID-19 infection or more than 1 year after the onset of infection. There should be a certain level of naturally improvement tendency at the earlier disease phase of long COVID, so the timing of treatment induction in the disease course would be very important.
Major Response 2: We thank the reviewer for the careful review of the manuscript. For 58 subjects, the period from a confirmed diagnosis of SARS-CoV-2 to the start of treatment was 113 ± 76.4 days on average, and the median was 81 days. We have added text to the Method section (Page 2, Lines 22-28).
Minor Point 1: In the Methods section, the authors described that the diagnosis method for COVID-19 just as “positive SARS-CoV-2 test”. To provide the information on the accuracy of the diagnosis for COVID-19, the authors should present whether the test is for the existence of SARS-CoV-2 antigen or SARS-CoV-2 mRNA by real time RT-PCR.
Minor Response 1: We agree with your assessment. The population for this study consisted of consecutive 58 COVID-19 patients with a confirmed diagnosis of SARS-CoV-2 by real-time reverse transcriptase-polymerase chain reaction (RT-PCR). We have added text to the Method section (Page 2, Lines 22-26).
Reviewer 2 Report
The authors showed that EAT was effective for ME/CSF symptoms among long COVID patients with chronic epipharyngitis. Treatments for ME/CSF condition have not been established yet, so the present study might have important role in the clinical practice setting amid COIVD-19.
I think however that there are a few improvements that should be made before publication.
Major Comments
None.
Minor Comments
( 1 ) The authors should declare the inclusion criteria of present study. The authors said that participants were long COVID patients who visited the specialty outpatient clinic for EAT. But there was no description for the numbers of total visited patients from August 2021 to November 2021, how many patients want to receive EAT, and among them how many patients the authors selected to present study. These points concern to selection biases. It is important points whether EAT is effective for ME/CSF long COVID patients without pharyngitis, so the authors should mention in the discussion section.
( 2 ) The authors should explain the concordance rate between the improvement ratio on inflammation grade of chronic pharyngitis and that on ME/CSF symptoms focusing on each case. This is an important step to discuss the mechanisms of EAT for ME/CSF conditions.
Author Response
Dear Reviewer 2
Thank you very much for reviewing our manuscript and offering valuable advice.
We have addressed your comments with point-by-point responses, and revised the manuscript accordingly.
Response to Reviewer 2 Comments
Point1: The authors should declare the inclusion criteria of present study. The authors said that participants were long COVID patients who visited the specialty outpatient clinic for EAT. But there was no description for the numbers of total visited patients from August 2021 to November 2021, how many patients want to receive EAT, and among them how many patients the authors selected to present study. These points concern to selection biases. It is important points whether EAT is effective for ME/CSF long COVID patients without pharyngitis, so the authors should mention in the discussion section.
Response 1: We thank the reviewer for the careful review of the manuscript. In this study, we performed endoscopy on all patients who visited the specialty outpatient clinic for Long COVID, not for EAT, and found that all had chronic epipharyngitis. All patients were treated with EAT after consenting to treatment. We excluded patients less than one month after the start of treatment, and the number of subjects was 58. Namely, the population for this study consisted of consecutive 58 COVID-19 patients who visited the specialty outpatient clinic for Long COVID. We have added text to the Methods, and Discussion sections (Page 2, Lines 22-30; Page 6, Lines 25-29; Page 7, Lines 26-27).
Point2: The authors should explain the concordance rate between the improvement ratio on inflammation grade of chronic pharyngitis and that on ME/CSF symptoms focusing on each case. This is an important step to discuss the mechanisms of EAT for ME/CSF conditions.
Response 2: Thank you very much for your excellent suggestion. Regarding your great advice, we investigated whether the reduction of chronic epipharyngitis affects the improvement of VAS score on each case. As a result, the reduction of chronic epipharyngitis by EAT contributed most strongly to the improvement of fatigue. Interestingly, even in patients whose inflammation did not improve, EAT alleviated the three symptoms of fatigue, headache, and attention disorder (Figure 4,5). These results indicate that EAT is effective at improving long COVID symptoms through a complex mechanism. We suggested that EAT improved Long COVID via the reduction of chronic epipharyngitis and neuromodulation by stimulation of the vagus nerve ending in the epipharynx. Actually, non-invasive vagus nerve stimulation (VNS) is considered to be useful to modulate peripheral and central inflammation response, headache, and mental distress associated with COVID-19. We have added text to the Introduction, Results, and Discussion sections (Page 2, Lines 14-16; Page 4, Lines 31- Page 5, Lines 7; Page 7, Lines 29-34; Page 7, Lines 41-49). In addition, we have changed Figure 4 and added Figure 5.
Reviewer 3 Report
Imai et al. showed that chronic epipharyngitis is present in patients with Long COVID, and EAT, a treatment for chronic epipharyngitis, may improve Long COVID symptoms.
This paper is attractive in that it provides a new perspective on Long COVID, where the etiology is unknown and a treatment has not been established.
To improve this paper, the Reviewer has the following suggestions:
- Missing link of chronic epipharyngitis in patients with Long COVID
Did everyone have subjective symptoms that suggest inflammation of the pharynx such as throat pain in the target patients? In other words, did this study target selected Long COVID patients who had pharyngeal symptoms and visited the specialty outpatient clinic for EAT?
If only some patients have subjective pharyngeal symptoms, the presence of chronic epipharyngitis may be a hidden focal inflammation in patients with Long COVID. This point is extremely important in this paper, and the authors need to clearly state it.
- Association between the severity of chronic epipharyngitis before treatment and the improvement of symptoms by EAT
Readers will be interested in the severity of chronic epipharyngitis before treatment and the degree of long COVID symptom improvement by EAT, which is missing from this paper. Thus, the Reviewer recommend that Figure 4 should be changed so that this point is clear.
Author Response
Dear Reviewer 3
Thank you very much for reviewing our manuscript and offering valuable advice.
We have addressed your comments with point-by-point responses and revised the manuscript accordingly.
Response to Reviewer 3 Comments
Point1: Missing link of chronic epipharyngitis in patients with Long COVID. Did everyone have subjective symptoms that suggest inflammation of the pharynx such as throat pain in the target patients? In other words, did this study target selected Long COVID patients who had pharyngeal symptoms and visited the specialty outpatient clinic for EAT? If only some patients have subjective pharyngeal symptoms, the presence of chronic epipharyngitis may be a hidden focal inflammation in patients with Long COVID. This point is extremely important in this paper, and the authors need to clearly state it.
Response 1: We thank the reviewer for the careful review of the manuscript. The population for this study consisted of consecutive 58 COVID-19 patients who visited the specialty outpatient clinic for Long COVID, not for EAT. Among the patients, 25 and 14 were aware of throat pain and post-nasal drip, respectively, that is, more than half of the patients had no throat symptoms. According to your excellent suggestion, we found that chronic epipharyngitis is present in patients with Long COVID regardless of the presence or absence of throat symptoms and may be involved in the pathophysiology of Long COVID. We have added text to the Methods, Discussion, and Conclusion sections (Page 2, Lines 22-30; Page 6, Lines 25-29; Page 8, Lines 9-10).
Point2: Association between the severity of chronic epipharyngitis before treatment and the improvement of symptoms by EAT. Readers will be interested in the severity of chronic epipharyngitis before treatment and the degree of long COVID symptom improvement by EAT, which is missing from this paper. Thus, the Reviewer recommend that Figure 4 should be changed so that this point is clear.
Response 2: Thank you very much for your excellent suggestion. Although there was no correlation between the severity of chronic epipharyngitis before treatment and the improvement of symptoms by EAT, we showed that EAT alleviated the three symptoms of fatigue, headache, and attention disorder, regardless of whether or not mucositis was reduced (Figure 4,5). These results suggest that the improvement of Long COVID is related to the various mechanisms of EAT, and we consider that not only the anti-inflammatory effect of EAT but also neuromodulation by stimulation of the vagus nerve ending in the epipharynxthe is effective. We have added text to the Introduction, Results, and Discussion sections (Page 2, Lines 14-18; Page 4, Lines 31- Page 5, Lines 7; Page 7, Lines 29-34; Page 7, Lines 41-49). In addition, we have changed Figure 4 and added Figure 5.